# Self-Care and Health-Related Quality of Life in Patients with Drainage Enterostomy: A Multicenter, Cross Sectional Study

**DOI:** 10.3390/ijerph18052443

**Published:** 2021-03-02

**Authors:** Eladio J. Collado-Boira, Francisco H. Machancoses, Ana Folch-Ayora, Pablo Salas-Medina, Mª Desamparados Bernat-Adell, Vicente Bernalte-Martí, Mª Dolores Temprado-Albalat

**Affiliations:** 1Department of Nursing, Faculty of Health Sciences, University of Jaume I, Castellón de la Plana, 12071 Castelló, Spain; colladoe@uji.es (E.J.C.-B.); afolch@uji.es (A.F.-A.); psalas@uji.es (P.S.-M.); bernatm@uji.es (M.D.B.-A.); bernalte@uji.es (V.B.-M.); 2Predepartmental Unit of Medicine, Faculty of Health Sciences, University of Jaume I, Castellón de la Plana, 12071 Castelló, Spain; 3Department of Nursing, University CEU Cardenal Herrera, Alfara del Patriarca, 46115 Valencia, Spain; maria.temprado@uch.ceu.es

**Keywords:** self-care, self-care monitoring, quality of life, stoma care

## Abstract

The current article examined stoma self-care and health-related quality of life in patients with drainage enterostomy, described clinical and sociodemographic variables and analyzed the relationship between all of them. Trained interviewers collected data using a standardized form that queried sociodemographic and clinical variables. In addition, Self-Care (SC) was measured through a specific questionnaire for Ostomized Patients (CAESPO) and Health-Related Quality of Life (HRQoL) through the Stoma Quality of Life questionnaire (S-QoL), which are not included in the electronic medical record. This was a multicenter, cross sectional study conducted in four hospitals of the province of Castellon (Spain), where 139 participants were studied. As novel findings, it was found that the level of SC of the stoma was high and was positively correlated with health-related quality of life. In relation to SC and sociodemographic variables studied in the research, women, married patients and active workers presented significantly higher scores than the rest. In relation to the clinical variables, we highlight the highest scores of the autonomous patients in the care of their stoma and those who used irrigations regularly. The lowest scores were the patients with complications in their stoma. We can highlight the validity and reliability of the CAESPO scale for biomedical and social research, and the importance of skills related to self-care of ostomy patients for a good level of HRQoL.

## 1. Introduction

Construction of a stoma is an integral part of many abdominal operations performed by general surgeons and others. Its use is common in the treatment of colorectal tumors, trauma, diverticulitis, inflammatory bowel disease and many other ailments. Colon cancer has become the main reason why patients end up carrying an elimination enterostomy [1]. Ostomies generate physical and psychological alterations that require significant adaptation by patients. Ostomized patients will have to modify their lifestyles and learn new skills related to the care of their stoma. This challenge can have a major impact on one’s-related quality of life (HRQoL). In chronic diseases such as the one we are dealing with, and in being a carrier of an elimination enterostomy, one of the health indicators that acquires special importance is the HRQoL, since it provides us with the perception that the patient has on his/her own health, that is, on his/her physical, psychological, and social limitations and on diminishing opportunities due to the disease, its sequelae and treatment [2]. The measurement of the HRQoL becomes the main indicator of health and adjustment to the disease in patients with a chronic illness or chronic condition such as being a carrier of an enterostomy [3,4].

Thus, patients ending up with a stoma must initially face serious surgery, the loss of important bodily function, a distortion of their image and a remarkable and important change in their physical functioning and personal care [5,6,7]. Hence, it is considered that all these changes affect various aspects of patients’ lives such as the psychosocial area, sexual health, body image or cultural and religious beliefs, among others [6,8,9,10,11,12,13]. In fact, these physical and psychological stressors play a very important role in the adjustment to the disease and in the prognosis of it, as well as in the quality of life perceived by the patient [5,14].

More than 65% of new Colorectal Cancer (CRC) cases occurred in countries with high or very high levels of human development; almost half of the estimated new cases occurred in Europe and the Americas [15]. In the United States in 2017 the annual age-standardized incidence rate for CRC was 40.7 per 100,000 inhabitants, and the mortality rate (2010–1014) was 14.8 per 100,000 inhabitants [16]. In Europe CRC is the second and the third leading cause of cancer death among men and women in Europe, with an incidence rate of 43.5 per 100,000 inhabitants and a mortality rate of 19.5 per 100,000 inhabitants, in the same period [17]. Incidence rates of CRC show high growth associated with economic development [18].

Together with the high incidence of CRC and other intestinal diseases that lead to intestinal surgery, this surgical technique is also greatly progressing. Thus, and with the aim of minimizing the side effects of this intervention, anterior resection with preservation of sphincter function is the treatment of choice [19]. Anastomotic leakage is one of the most feared complications after CRC surgery. It occurs in approximately 5% of patients following potentially curative resection and is more common in those undergoing rectal surgery and those with a low anastomosis [20]. It has been observed that the completion of a temporary stoma is effective in reducing anastomotic leakage rates and reoperation rates after previous resection in patients with rectal cancer [19,20]. In those cases, the patients lose the function of the sphincter (some permanently and others temporarily). However, in those cases, patients experience a strong change in their lives that translates into a great psychological impact.

Within the areas of life that are affected in enterostomy carriers and that seriously affect their perception of quality of life, these patients also require an important behavioral change in daily routines, as well as inclusion in life daily of behaviors towards personal care and hygiene that require a lot of time and dedication [21,22,23].

The first personal behavioral change caused by enterostomy is that the patient requires knowledge and psychomotor skills to adequately manage the stoma [23,24,25,26]. Changes such as the handling and emptying of an ostomy bag system, monitoring and caring for irritations in the peristomal area and learning how different foods and beverages affect the consistency and volume of fecal effluent. The patient’s ability to care for their ostomy is termed self-care (SC), defined by Orem, Taylor and Renpenning as “the practice of learned actions, directed towards oneself or towards the environment, to regulate the factors that affect their own functioning or development for the benefit of their life, health or well-being” [27]. SC is considered essential in the management of chronic illness [28] and has been considered as a possible way to achieve positive health outcomes related to psychological adjustment in patients with a colostomy [29]. A successful adjustment to the permanent colostomy is more probably when the patient is adequately instructed in self-care [30].

With the referenced antecedents, we consider it of great interest to investigate the relationships between Stoma SC, clinical and sociodemographic variables and HRQoL of ostomates. These aspects can be essential for the nursing professionals responsible for caring for these patients.

## 2. Materials and Methods

We designed a multicenter, cross sectional study conducted in four hospitals of the province of Castellon (Spain) with a colorectal surgery service and a stoma nursing unit. The study was classified by the Spanish Agency for Medicines and Healthcare Products, which belongs to the Ministry of Healthcare, Social Services and Equality, as a “Nonpost authorization observational study” with code number 2828/RG5549, and authorized by the Ethical committee for clinical research of the Castellón Provincial Hospital Consortium, with authorization date 22 February 2012. All the participating (N = 139) were extracted from these 4 hospitals, with special care to guarantee their privacy. They were assigned an identification number in the study that prevented their identification. All participants gave their informed consent before taking part in the study.The primary purpose of this study was to assess stoma SC and HRQoL in patients with drainage enterostomy, describe clinical and sociodemographic variables and analyze the correlations between all of them.

We hypothesized that:Stoma SC in patients with drainage enterostomy would be poor.Some domains of both stoma SC would be associated with HRQoL in patients with drainage enterostomy.Some SC ostomy-related sociodemographic (sex, age, number of children, education level and marital status) and clinical characteristics would be correlated with stoma SC in patients with drainage enterostomy.

To estimate the sample size, we relied on Liu and colleagues [31], who reported a 19% incidence of stoma and peristoma complications in a group of stomized patients. To estimate a proportion using a normal asymptotic confidence interval with a two-sided 95% correction for finite populations setting the precision at 3.5%, a minimum sample size of 117 subjects is estimated. Considering a 10% dropout rate, 139 subjects were recruited. All the computations for the estimation of this sample size have been carried out with the software Sample Size Estimation Software, version 3.0 (Vanderbilt University Biostatistics, Nashville, Tennessee).

Inclusion criteria were (a) colostomy or ileostomy performed at least 3 months before inclusion in the study, (b) over 18 years of age, (c) having spoken communication skills, (d) written communication and (e) having fluency in Spanish, and (f) being able to give informed consent. Those patients undergoing stoma surgery as palliative care were excluded. Table 1 indicates the characteristics of the hospitals included in the study, in which 185 ostomized patients were identified and contacted for their participation. Of these 185 subjects, 32.4% (*n* = 60) did not meet the inclusion/exclusion criteria; the remaining 67.6% (*n* = 125) gave their consent to participate.

The protocol, which was developed by a multidisciplinary team that included nurses and physicians, was created for the use of critical care RNs practicing in the colostomy office by 3 interviewers who were trained to homogenize the data recollection of the CAESPO and S-QoL (Spanish adaptation) [32] questionnaires. A common problem when using different interviewers is that each one has a style of writing or collecting information. To avoid this bias, a series of training sessions were carried out to homogenize their information collection styles. The Jaume I University of Castellón (Spain) coordinated the administration of the questionnaires and instruments, as well as the collection of information. Data were collected from January 2016 to January 2017. The information was extracted and encoded in an SPSS matrix using the identification code that anonymizes the responses of the subjects. Other relevant clinical information was obtained from electronic medical record review.

The variables collected were the following:

### 2.1. Socio-Demographic and Medical Characteristics

Sociodemographic characteristics included sex, age, number of children, education level and marital status. The clinical characteristics of the ostomy consulted in the medical history included: type (ileocolostomy or colostomy), date of intervention, diagnosis justified the ostomy, temporality of the ostomy (temporary or permanent), location of the stoma, use of irrigation, stool characteristics, type of ostomy bag and autonomy in the ostomy care.

### 2.2. The Specific Self-Care for Ostomized Patients Questionnaire (CAESPO)

The CAESPO [32] assesses 3 dimensions: 21 items evaluate “general SC”, 18 items query “personal development and social interaction SC (development SC)”, and 18 items query aspects of “specific SC related to the presence of an ostomy (specific SC)”. Each subscale assesses knowledge, practice and interest in SC, as indicated in Orem’s theory of the 3 SC domains (knowledge of SC, interest and attitude toward SC, and SC behaviors). Items are scored using a 4 points scale (low level to high level of SC). The CAESPO direct score ranges from 58 to 232, becoming a scale from 0 to 100 where higher scores indicate higher levels of self-care.

### 2.3. The Stoma Quality of Life (S-QoL)

The S-QoL [33] contains 2 factors—stoma care subscale and social subscale—covered in 28 items. Respondents choose 1 of the 5 possible answers, ranging from “not at all confident” to “extremely confident”. Higher scores correspond to higher levels of confidence.

### 2.4. Validity and Reliability/Rigour

All the measuring instruments used have shown adequate reliability and validity. More specifically, the internal consistency of the CAESPO score was good (α = 0.889) and Test-retest reliability was excellent (α = 0.987); the Construct Validity indices obtained by SEM are adequate (χ^2^ = 43.132, *p* < 0.001; RMSEA = 0.155 [0.107–0.204]; BBNFI = 0.957; CFI = 0.967; IFI = 0.968). Cronbach α values for the CAESPO subscales were: general SC (α = 0.754), development SC (α = 0.786), and specific SC (α = 0.908) [31]; and Cronbach α values for the S-QoL total score, stoma care SE, and social SE were 0.97, 0.97, and 0.89 [33].

Data were analyzed using the Statistical Package for the Social Sciences software version 23.0 (SPSS, Chicago, IL, USA). All statistical tests were 2-tailed and the threshold for statistical significance was an α level of less than 0.05. We first performed the corresponding descriptive statistical analyses, including the mean (standard deviation, SD) and the frequency (percentage), to summarize and present the data. On the other hand, we examine all the variables to determine their assumption of normality using the Kolmogorov-Smirnov test. We then used Spearman’s correlation analysis to investigate the correlation between each subscale of CAESPO and S-QoL, and due to their non-normality. Lastly, we used independent Mann–Whitney U and Kruskal–Wallis tests to explore the differences in the scores of both questionnaires and their subscales according to sociodemographic or clinical data.

## 3. Results

Of the total number of 139 subjects, 14 subjects did not complete the questionnaires and 5 were eliminated for having extreme values, resulting in the final sample of 120 subjects. The mean age of the 120 subjects taking part in this study was 66.91 years (SD: 11.82), and the mean of days since the placement of the stoma was 1379.9 days (SD: 1227.32). Most of the participants were males (64.2%, *n*: 77), married (83.3%, *n*: 100), retired (75.8%, *n*: 91), autonomous in their care (79.8%, *n*: 95), with primary studies (45%, *n*: 54) and with an annual income of less than 6.000€ (64.17%, *n*: 77). With regard to the clinical characteristics of the participants, most reported no stoma-related complications (84.7%, *n*: 100), the majority of them being colostomies (81.7%, *n*: 98), permanent (65%, *n*: 78), and, in most cases, using one-piece closed collection systems (40.8%, *n*: 49). A more detailed breakdown of these socio-demographic variables can be observed in Table 2.

Table 3 presents the average scores obtained in the different subscales of the instruments used, together with their normality tests. It is observed that all scores, both average and median, of the subscales yielded values above 50%. The only scores that followed a normal distribution were STOMA quality of life global score and STOMA quality of life personal subscale (marked with “†” in Table 3).

Given the non-normality of the scores, a Spearman’s Rho correlation analysis was carried out between the scores of the different dimensions of the instruments used, observing that there was a significant correlation between the STOMA quality of life scores among the general SC knowledge and general SC practice (Table 4).

Regarding the sociodemographic variables, significant differences were found according to sex, based on the Mann–Whitney U test, in the CAESPO general SC score (U_M-W_ = 32,227.50, *p* = 0.043), with the highest score being observed in women. No differences according to sex were observed for any of the other variables (*p* > 0.05).

In relation to marital status, and according to the Kruskal–Wallis test, significant differences were found for the CAESPO development SC score (χ^2^ = 12.022, *p* = 0.007), and for the social dimension of Stoma QoL (χ^2^ = 11.292, *p* < 0.010), where married participants show a higher score than the rest. Significant differences according to work activity were also observed for CAESPO development SC scores (χ^2^ = 10.986, *p* < 0.010), and the CAESPO specific SC scores (χ^2^ = 23.173, *p* < 0.010), with those employed obtaining higher scores. No significant differences were observed for any of the SC scores studied according to the level of studies across the two instruments (*p* > 0.05).

Regarding the clinical variables, differences between groups were observed according to the autonomy of care, the presence of complications in the stoma, the use of irrigation and the type of effluent. According to the autonomy of the subjects in relation to the care and management of the stoma, there were significant differences in all self-care factors, CAESPO general SC scores (U_M-W_ = 861.50, Z = −2.343, *p* = 0.011), CAESPO developmental SC scores (U_M-W_ = 800.5, Z = −2.839, *p* < 0.010) and, CAESPO specific SC scores (U_M-W_ = 809.50, Z = −2.668, *p* = 0.008), with those subject who are autonomous in their care obtaining higher scores.

Considering the complications of the stoma, significant differences in the Stoma QoL Personal factor scores (U_M-W_ = 737, Z = −2.062, *p* = 0.039), and in the Stoma QoL Social factor scores (U_M-W_ = 532.5, Z = −3.534, *p* < 0.01) were found. Furthermore, with respect to CAESPO, significant differences with respect to the general SC scores (U_M-W_ = 728.5, Z = −2.166, *p* = 0.030) were found, with those patients with complications in the stoma obtaining a worse quality of life and a worse level of self-care.

Regarding the use of irrigations through the stoma as a control measure of intestinal effluents, differences in the level of self-care measured with CAESPO were observed, both in the developmental SC score (U_M-W_ = 655.5, Z = −2.885, *p* = 0.004), and in the specific SC score (U_M-W_ = 614, Z = −3.167, *p* = 0.002), with patients who perform irrigation obtaining a higher level of self-care.

In relation to Effluents, and according to self-care and CAESPO scores, significant differences with respect to general SC Score (H_K-W_ = 10.432, *p* = 0.015) were obtained, not finding differences in Effluents for both the developmental SC score (H_K-W_ = 1.605, *p* = 0.658), and the specific SC score (H_K-W_ = 1.938, *p* = 0.585).

No significant differences were observed (*p* > 0.05) between groups considering the following variables: type of ostomy, temporality of the stoma or the type of drainage device used.

## 4. Discussion

First, we hypothesized that Stoma SC in patients with drainage enterostomy would be poor. Our study approaches the problem, in an original and novel way, from a nursing theoretical framework based on the self-care model. Given that it is the first time that the Specific CAESPO self-care questionnaire has been used, it is not possible to compare it with other samples or to determine the exact level of self-care of the studied population. However, given that the different factors and dimensions of self-care yielded scores that exceeded, in general, the average established in the validation of the instrument [33], it can be considered that the level of acquisition of competences in self-care by the studied population was high; although, it could be improved in some aspects.

The level of self-care in patients with ostomies has been studied, using other instruments, in previous studies. Thus, it is noteworthy that the present study’s results are similar and comparable to those of Knowles and colleagues [34] with a sample of colostomized Australian patients and to those obtained by Wu et al. [35], in a sample of 96 stoma patients in Hong Kong. As in our results, both studies show raised levels of self-care in ostomized patients. However, our results differ from those presented by Su and colleagues [36], which were considerably poorer, perhaps conditioned by being exclusively patients with temporary ostomies. Other studies have shown the temporality of the type of stoma as a conditioning factor, compared to stoma acceptance and stoma care self-efficacy [7]; although, our results do not indicate significant differences in the stoma temporality.

On the other hand, we hypothesized that stoma SC would be associated with HRQOL in patients with drainage enterostomy. Our analyses show that self-care has a direct relationship on the well-being and health of patients with an enterostomy, revealing self-care as a predictor of quality of life, both in relation to social and personal factors. This information is in clear consonance with the conclusions of other research conducted on self-care and quality of life in ostomized patients [29,30,35,37,38,39,40]. The same phenomenon has been previously analyzed in other chronic diseases [41].

Lastly, we investigated about some SC ostomy-related sociodemographic (sex, age, number of children, education level and marital status) and clinical characteristics would be correlated with stoma SC in patients with drainage enterostomy.

Female patients scored higher on the General SC Knowledge subscale than male patients. Similarly, patients who live as a couple also showed significantly higher values in the CAESPO Developmental SC Practice, Stoma QoL and Social dimension of Stoma QoL compared to those who do not. Our results contrast with previously published studies that have found that women and married patients can deal better with CRC than men and unmarried patients [36,42]. Probably, the average age of the sample and sociocultural factors, such as employment activity, can justify this association.

In relation to the clinical variables, we found that patients with some type of complication related to the stoma showed differences in the level of quality of life, which was reduced. This association coincides with the findings of other studies [4,35,36] and highlights the great importance of the stoma therapist’s role after colorectal surgery in the therapeutic education of these patients. It is essential to acquire knowledge and skills for the adequate care of the periostomal skin, the use of drainage devices and the management of possible complications, which will derive in greater autonomy [43]. Our findings coincide with those of other authors [29], in which those patients who are autonomous and can clean and change their ostomy pouch by themselves present a higher level of self-care and a higher quality of life.

Other clinical variables that have been shown to be associated with SC are irrigations and type of effluent. Colostomy irrigation involves irrigating the colon regularly to establish a regular bowel habit [44]. Other previous studies have already inquired about the role of colostomy irrigation in the quality of life of patients [45,46]. In our patients, those patients who used irrigation as a method of controlling fecal waste, showed a higher level of significant self-care in the CAESPO development and specific score, showing greater involvement in all aspects related to the control of their process, although they did not show a higher level of quality of life than the rest.

Contrary to the findings of other authors [40,46], we have not found differences in the level of self-care or quality of life depending on the type of ostomy. However, we have found differences according to the type of effluent. The stool of patients with ileostomy is usually pasty or even fluid, which facilitates the appearance of periostomal lesions due to the moisture over the skin and the acidity of the stool [47]. We have found a higher level of self-care among those patients with stool with pasty consistency versus those with normal or hard consistencies. Similarly, these patients have shown higher levels of quality of life, which is a finding consistent with Magistri et al.’s results [46].

From the findings presented, it is necessary to highlight the relationship between the level of self-care and the quality of life of ostomized patients. In this aspect, the role of the stoma therapist is fundamental. It is necessary to consider in the care of these patients that the influence of some clinical variables such as complications related to the stoma or the characteristics of the stool, as predictors of a poor quality of life.

## 5. Strengths and Limitations

The main bias in our research was sample selection bias. Due to the fragility of this sample due to the characteristics of the patients themselves (the majority with colon cancer), access was difficult and its randomization could not be considered, so this study was carried out in an observational way and without performing any intervention educational or health. However, the multicenter design in four hospitals allowed us to have a large sample size and an overview of the state of the art. All this makes this study a consistent and representative study. Lastly, it should be noted that the strict inclusion/exclusion criteria led to a high number of patients (32.4%) being excluded from the study.

## 6. Implications

There are several tools in the scientific literature to measure specific self-care in patients with an ostomy. This research is the first in which the CAESPO scale has been used to measure self-care competencies and the results have been used to demonstrate its influence on health-related quality of life. This study shows the scientific community the validity and reliability of this tool for biomedical and social research.

## 7. Conclusions

In the present study, we attempted to examine the level of SC and its relation to QoL in a sample of 120 Spanish patients with an elimination enterostomy.

We found that the stoma SC level was high and was positively correlated with the QoL of these patients. Sociodemographic variables including gender, marital status and economic income were also associated with the level of stoma SC. The clinical variables associated with the level of SC were the presence of complications related to the stoma, the type of effluent and the use (or not) of regular irrigations of the colon.

Based on the results of this study, we can highlight the great importance that the acquisition of a high level of skill related to SC by ostomized patients has for a good level of HRQoL. The health services must provide patients with the necessary tools for their acquisition.

## Figures and Tables

**Table 1 ijerph-18-02443-t001:** Characteristics of the hospitals included in the study.

	Beds	Surgical Procedures Completed in 2015
General Hospital of Castellon	574	12.529
Provincial Hospital of Castellón	263	6.702
Vila-Real Hospital	258	8.764
Vinaroz Hospital	139	4.986

**Table 2 ijerph-18-02443-t002:** Characteristics of the subjects included in the study.

**Socio-Demographic**		
Mean age, in years ± SD		66.91 ± 11.82
Median n° children (min, max)		2 (0, 7)
Sex	Women	43 (35.8%)
Marital Status	Single	6 (5%)
	Married	100 (83.3%)
	Separated—Divorced	2 (1.7%)
	Widow/er	12 (10%)
Educational level	None	33 (27.5%)
	Basic	54 (45%)
	Secondary school—Voc. Tr.	22 (18.3%)
	University	11 (9.2%)
Employment Status	Retired	91 (75.8%)
	Employee/Self-employed	14 (11.7%)
	Others	15 (12.5%)
Clinical		
Mean n° days with the Stoma		1379.9 ± 1227.32
Diagnosis	Colon cancer	104 (86.7%)
	Crohn	8 (6.7)
	Diverticulitis	3 (2.5%)
	Other	5 (4.2%)
Type of ostomy	Colostomy	98 (81.7%)
	Ileostomy	22 (18.3%)
Autonomous in their care	Yes	95 (79.8%)
Complications	No Complications	100 (84.7%)
	Stenosis	11 (9.3%)
	Oedema	1 (0.8%)
	Prolapse	3 (2.5%)
	Retraction	1 (0.8%)
	Pain	2 (1.7%)
Type of ostomy bag	One Piece bag	49 (40.8%)
	Drainable One Piece	26 (21.7%)
	Two Pieces bag	23 (19.2%)
	Drainable Two Pieces	22 (18.3%)
Appliance time frame	Permanent	78 (65%)
Stoma level	Invaginated	7 (5.8%)
	Flush	48 (40%)
	Protruding	65 (54.2%)
Use of irrigations	Yes	21 (16.8%)
Stool Characteristics	Liquid stool	13 (10.4%)
	Pasty	59 (47.2%)
	Normal	45 (36%)
	Hard	8 (6.4%)

**Table 3 ijerph-18-02443-t003:** Descriptive statistics of Questionnaires.

	Mean	Median	SD	Min	Max	K-S	*p*
CAESPO General Self Care	70.82	72.77	9.42	45.56	92.78	0.134	0.000
CAESPO Developmental Self Care	74.32	73.88	7.96	46.39	94.72	0.128	0.000
CAESPO Specific Self Care	74.42	72.91	10.91	46.34	100	0.128	0.000
S-QoL Global	80.94	72.5	12.46	51.25	100	0.079	0.066 ^†^
S-QoL F1 (Personal)	73.85	90	15.67	40	100	0.081	0.053 ^†^
S-QoL F2 (Social)	88.00	66.67	12.38	47.50	100	0.166	0.000

CAESPO: The Specific Self-Care for Ostomized Patients Questionnaire. S-QoL: The Stoma Quality of Life Questionnaire. K-S: Kolmogorov-Smirnov Test for Normality [^†^: variable follows normal distribution].

**Table 4 ijerph-18-02443-t004:** Spearman Rho (ρ) Correlation CAESPO—STOMA QoL.

		Knowledge—SQoL	Practice—SQoL
CAESPO General Score	ρ	0.227 *	0.366 **
	*p*	0.011	0.000
	N	124	125
CAESPO Developmental Score	ρ	−0.051	0.067
	*p*	0.572	0.456
	N	124	125
CAESPO Specific Score	ρ	−0.059	−0.006
	*p*	0.519	0.949
	N	124	125

S-QoL: The Stoma Quality of Life Questionnaire. *: *p* < 0.05; **: *p* < 0.01.

## Data Availability

The data presented in this study are available on request from the corresponding author, upon reasonable request. The data are not publicly available due to their containing information that could compromise the privacy of research participants.

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
