# Peer review of "Self-Care and Health-Related Quality of Life in Patients with Drainage Enterostomy: A Multicenter, Cross Sectional Study"

_ijerph, 2021, doi:10.3390/ijerph18052443_

Round 1

Reviewer 1 Report

The authors have described a study on Self-care and Health-related Quality of Life in patients with drainage enterostomy. 

I would suggest highlighting the novelty of the study as the current format does not appear to represent significant new information to existing knowledge. 

I would suggest clarifying the results in the abstracts section  (see below) as these statements merely point out differences and do not seem to clearly show what the novel findings are 

"Significant differences were found in Self Care according to sex (p = .043); married participants show higher score (p < .01); also, significant differences according to work activity were observed (p < .01). Regarding the clinical variables, differences were observed according to the autonomy of care, the presence of complications in the stoma, the use of irrigation and the type of effluent (p < .01). "

I would suggest adding more details on the studied aspects of QOL. There are two major variables investigated "CAESPO" and "S-QOL". However, details and rationale for using them and are very limited within the text.

The authors have hypothesized that "Stoma SC in patients with drainage enterostomy would be poor". As mentioned in the manuscript this could be dependent on different variables including the patient's education on self-care. I don't believe that poor stoma SC is applicable to all CRC affected patients.

The authors hypothesized that "Some domains of both stoma SC would be associated with HRQOL in patients with drainage enterostomy". These domains need to be defined as described in details.

The authors hypothesized that "Some ostomy-related sociodemographic and clinical characteristics would be correlated with stoma SC in patients with drainage enterostomy". I would suggest clearly defining "ostomy-related sociodemographic" as the following variables described in methods  (Sociodemographic characteristics included sex, age, education level, marital status, place of residence, living status, primary caregiver, occupation, and monthly family income ) do not seem to be ostomy specific. 

Author Response

First of all, we want to acknowledge the contributions and comments of all reviewers. We believe that we have been able to take them all into account, which has substantially improved the article we present. Next, we will comment on each of the points indicated by your review.

I would suggest clarifying the results in the abstracts section  (see below) as these statements merely point out differences and do not seem to clearly show what the novel findings are 

"Significant differences were found in Self Care according to sex (p = .043); married participants show higher score (p < .01); also, significant differences according to work activity were observed (p < .01). Regarding the clinical variables, differences were observed according to the autonomy of care, the presence of complications in the stoma, the use of irrigation and the type of effluent (p < .01). "

A redrafting of the abstract has been carried out, including the suggestions of several reviewers. We hope it conforms more to the indicated indications.

I would suggest adding more details on the studied aspects of QOL. There are two major variables investigated "CAESPO" and "S-QOL". However, details and rationale for using them and are very limited within the text.

The introduction has been modified in order to present in a more rational way the details for the use of CAESPO and S-QoL, as well as a series of clinical variables has been modified and introduced in the descriptive ones, as indicated by more reviewers.

The authors have hypothesized that "Stoma SC in patients with drainage enterostomy would be poor". As mentioned in the manuscript this could be dependent on different variables including the patient's education on self-care. I don't believe that poor stoma SC is applicable to all CRC affected patients.

Completely agree. The results of the study cannot be generalized to all patients with CRC. We have indicated more clearly that the hypotheses of the study only refer to patients with CRC who have an ostomy, where this situation is more frequent.

The authors hypothesized that "Some domains of both stoma SC would be associated with HRQOL in patients with drainage enterostomy". These domains need to be defined as described in detail.

The domains related to Stoma SC and HRQOL are widely described in the section that describes the measurement tools of the study variables. We consider that this location of the detailed description is more practical than elsewhere in the paper.

The authors hypothesized that "Some ostomy-related sociodemographic and clinical characteristics would be correlated with stoma SC in patients with drainage enterostomy". I would suggest clearly defining "ostomy-related sociodemographic" as the following variables described in methods  (Sociodemographic characteristics included sex, age, education level, marital status, place of residence, living status, primary caregiver, occupation, and monthly family income ) do not seem to be ostomy specific. 

You are correct, we have clarified that we are referring to self-care variables (SC) related to ostomies, and we have described them as you have indicated.

A quick review of the English wording has been carried out, but with the deadline provided it couldn’t be more in-depth. The new submitted article is still under review so that, in the case of acceptance, carry out the last modifications if they are necessary.

Reviewer 2 Report

Collado-Boira et al. examined stoma self-care and health-related quality of life (HRQL) in patients with drainage enterostomy that describe clinical and sociodemographic variables and analyze their relations. This is an important and yet less focused area of research because it really impacts patient HRQL. The study was a multicenter, cross sectional study conducted in four hospitals of the province of Castellon, Spain. The study was carried on for one-year 2016/2017 that included 120 subjects undergoing stoma surgery as palliative care allowing data collection using a standardized form that queried sociodemographic and clinical variables, collected from the electronic medical record (chart review), in addition to the Specific Self-Care for Ostomized Patients Questionnaire (CAESPO) and Stoma Quality of life (S-QoL). The study highlights the importance of the skills related to self-care by ostomized patients has for a significant beneficial level of HRQL. The health services must provide patients with the necessary tools for their acquisition and stoma therapist plays a very important role.

The paper is of informative nature, attractive and important to health professionals dealing with patients with ostomies. The data for characteristics of the hospitals included in the study (table 1), the subjects (table 2) descriptive statistics of questionnaires (table 3), and Spearman Correlation (table 4) are well presented and easy to follow. The 47 references correspond well with the article’s presentation.

Author Response

First of all, we want to acknowledge the contributions and comments of all reviewers. We believe that we have been able to take them all into account, which has substantially improved the article we present.

A quick review of the English wording has been carried out, but with the deadline provided it couldn’t be more in-depth. The new submitted article is still under review so that, in the case of acceptance, carry out the last modifications if they are necessary.

Reviewer 3 Report

General

Authors explore in their manuscript ‘Self-care and Health related Quality of Life in patients with drainage enterostomy: a multicenter, cross sectional study’ self-care and hrqol in Spanish pts with a stoma. Such features are understudied, however some major work should be done before this manuscript can be published.

Title

-

Abstract

Background

-

Methods

Start this part with the sentence:

<We designed a multicenter, cross sectional study conducted in four hospitals of the province of Castellon (Spain) where 120 participants were studied.>

<Trained interviewers> - what did these interviewers do, and in what did you train them?

Were CAESPO and S-QoL already in the electronic medical record? It is not clear to me.

Results

-

Conclusion

<We can highlight the importance of the skills related to self-care by ostomized patients has for a good level of quality of life related to health.> - it is not clear to me how QoL plays a role in this, and why it is important, as it is not at all mentioned in the Results section.

Key words

Consider deleting <Health promotion, Nurse – Patient interaction> - these words don't pop up in the title of the manuscript, nor in the Abstract

Introduction

<your health related quality of life> - one’s (HRQoL)

<patients with a stoma must face serious surgery> - patients ending up with a stoma must initially face serious surgery

<In the United States in 2017, there were projections of 135,430 individuals being newly diagnosed with CRC and 50,260 patients dying from the disease (13).> - I have some questions about this:

  • Were the projections restricted to the USA, or are these global projections?
  • When will this happen?

In the paragraph starting with <In chronic diseases> the role HRQoL plays is not clear; especially the last sentence is not at all clear. Why is this para added?

Methods

Sample

The first part what is now under Methods (l 101-109) should be moved up, after the aim.

Could you tell me why you submitted this manuscript (let’s say Jan 2021) so long after data collection (ended in Jan 2017).

Measures

<Sociodemographic characteristics included sex, age, education level, marital status, place of residence, living status, primary caregiver, occupation, and monthly family income.> please replace this sentence by <Sociodemographic characteristics included sex, age, number of children, education level, marital status, place of residence, living status, primary caregiver, occupation, and monthly family income.> as it is like this in the table.

<stoma care subescale and social subescale>

<in the original study (32).> - how were these outcomes in your sample (thus not in the original study)?? And how did you translate this questionnaire?

Statistical analyses

Please rewrite this section: First, we … . Then, we … . Next, we … . Finally, we … . The readership easier grabs what you did and in which order the Results will be shown.

Results

<with "*" in Table 2).> - replace the * by another sign (also in the Table) (* stands for significant); and make of it <in Table 3).>

Why no Table 5? This table could be the most important! self-care and quality of life

Discussion

Please keep in mind the following structure for writing a Discussion:

para1                    start with repeating the research question + answer this without any comments or interpretation.

Para2,3,#              start a new para, 1 topic per para, and start this para with one of your findings – which then defines the content of the para. Relate your finding to earlier published references. Close this para by stating how existing theory changed because of your finding.

Strengths and limitations

Mention Strengths! Limitations are those issues, which might bias your findings (e.g. are there sources of bias, like information bias or selection bias and how did you handle confounders?). Include the direction in which they did. Research not yet carried out is not a limitation, but should be brought to the next chapter.

Implications

(split into: for practice/policy, for future research)

Conclusion

Short answer to the Aim; + your most important implication; as it is it’s ok – except the last sentence <in this learning process, the figure of the stoma therapist plays a very important role.>; it should disappear.

The Discussion section contains a discussion of the findings; new findings cannot be present in the Discussion section.

Please rewrite the Discussion section

Tables, Figures

T2 replace <Estudios> by Educational level

Clarify <Piece with no tap> <Piece with a tap> <Pieces with no tap> <Pieces with a tap>; I don't understand

<in Table 3).>

Why no Table 5?

Author Response

First of all, we want to acknowledge the contributions and comments of all reviewers. We believe that we have been able to take them all into account, which has substantially improved the article we present. Next, we will comment on each of the points indicated by your review.

Methods

Start this part with the sentence:

<We designed a multicenter, cross sectional study conducted in four hospitals of the province of Castellon (Spain) where 120 participants were studied.>

We have modified the beginning of this section as you have indicated.

<Trained interviewers> - what did these interviewers do, and in what did you train them?

The interviews were conducted by 3 different researchers. A common problem when using different interviewers is that each one has a style of writing or collecting information. To avoid this bias, a series of training sessions were carried out to homogenize their information collection styles, of course prior to the beginning of the research.

Were CAESPO and S-QoL already in the electronic medical record? It is not clear to me.

Information on CAESPO AND S-QOL were obtained in the interviews. The clinical variables for diagnosis and treatment were obtained from the electronic medical record. We have included this clarification when we talk about the interviewers' training sessions, indicating that their task consisted of passing the CAESPO and S-QoL questionnaires among other tasks.

Conclusion

<We can highlight the importance of the skills related to self-care by ostomized patients has for a good level of quality of life related to health.> - it is not clear to me how QoL plays a role in this, and why it is important, as it is not at all mentioned in the Results section.

We justify this statement by the results shown in table 4. The Spearman's Rho correlation analysis indicates a relationship between the scores on the self-care scale and STOMA quality of life scores. Our analyses show that self-care has a direct relationship on the well-being and health of patients with an enterostomy, revealing self-care as a predictor of quality of life, both in relation to social and personal factors. We have modified this table to make these questions clearer.

Key words

Consider deleting <Health promotion, Nurse – Patient interaction> - these words don't pop up in the title of the manuscript, nor in the Abstract

We have considered and we have deleted it.

Introduction

<your health related quality of life> - one’s (HRQoL)

We have modified all the acronyms in the text so that they have the same structure (HRQoL and QoL)

<patients with a stoma must face serious surgery> - patients ending up with a stoma must initially face serious surgery

We have modified that sentence so that it is worded as you have indicated.

<In the United States in 2017, there were projections of 135,430 individuals being newly diagnosed with CRC and 50,260 patients dying from the disease (13).> - I have some questions about this:

Were the projections restricted to the USA, or are these global projections?

When will this happen?

Those projections were focused on the US and, as you seems to indicate, it would be too restricted a projection. That is why the data found for Europe has also been included. We add: In Europe CRC is the second and the third leading cause of cancer death among men and women in Europe, with more than 242,000 deaths estimated in 2018 (17).

In the paragraph starting with <In chronic diseases> the role HRQoL plays is not clear; especially the last sentence is not at all clear. Why is this para added?

At the direction of other reviewers, we have relocated this paragraph to the beginning of the introduction, where we believe the role of HRQoL is clearer.

Methods

Sample

The first part what is now under Methods (l 101-109) should be moved up, after the aim.

Your directions have been followed, and that part has been relocated.

Could you tell me why you submitted this manuscript (let’s say Jan 2021) so long after data collection (ended in Jan 2017).

This article is the last in a series of reports published on this research since 2017. This, together with two previous failed publication attempts, have delayed its possible publication.

However, we consider that the interest of the results justifies its publication despite its chronology.

Measures

<Sociodemographic characteristics included sex, age, education level, marital status, place of residence, living status, primary caregiver, occupation, and monthly family income.> please replace this sentence by <Sociodemographic characteristics included sex, age, number of children, education level, marital status, place of residence, living status, primary caregiver, occupation, and monthly family income.> as it is like this in the table.

The text has been modified to adapt to the data presented in the table.

<stoma care subescale and social subescale>

It has been modified in all the text: "subescale" by "subscale"

<in the original study (32).> - how were these outcomes in your sample (thus not in the original study)?? And how did you translate this questionnaire?

The reliability and validity data of the original study are those exposed in this article, just in the same paragraph. The methodology used is published in: Collado-Boira, E. J., Machancoses, F. H., & Temprado, M. D. (2018). Development and validation of an instrument measuring self-care in persons with a fecal ostomy. Journal of Wound Ostomy & Continence Nursing, 45(4), 335-340.

This questionnaire has been created in Spanish and applied to native Spanish subjects. Various translation proposals are being studied and their subsequent validation into other languages, including English and Chinese. For this we are consulting with experts in the field native to those languages, and with a high knowledge of Spanish. Validation studies in other languages are expected to be carried out shortly, but the COVID19 situation has changed all plans.

Statistical analyses

Please rewrite this section: First, we … . Then, we … . Next, we … . Finally, we … . The readership easier grabs what you did and in which order the Results will be shown.

It is much clearer that way. Thank you very much for your indication.

Results

<with "*" in Table 2).> - replace the * by another sign (also in the Table) (* stands for significant); and make of it <in Table 3).>

We have used the symbol "†".

Why no Table 5? This table could be the most important! self-care and quality of life

The relationship between SC and QoL is represented in Table 4, in which the spearman correlation has been carried out between the Caespo (SC) dimensions and the STOMA QoL score. We have modified the table to make these questions much clearer.

Discussion

Please keep in mind the following structure for writing a Discussion:

para1                    start with repeating the research question + answer this without any comments or interpretation.

Para2,3,#              start a new para, 1 topic per para, and start this para with one of your findings – which then defines the content of the para. Relate your finding to earlier published references. Close this para by stating how existing theory changed because of your finding.

We greatly appreciate the comments provided. We have proceeded to restructure the discussion so that it is more clarifying. Similarly, we have highlighted the main findings of our research.

Strengths and limitations

Mention Strengths! Limitations are those issues, which might bias your findings (e.g. are there sources of bias, like information bias or selection bias and how did you handle confounders?). Include the direction in which they did. Research not yet carried out is not a limitation, but should be brought to the next chapter.

Implications

(split into: for practice/policy, for future research)

We have added the sections "Strenghts and Limitations" and "Implications" following your instructions.

Conclusion

Short answer to the Aim; + your most important implication; as it is it’s ok – except the last sentence <in this learning process, the figure of the stoma therapist plays a very important role.>; it should disappear.

Please rewrite the Discussion section

The Discussion section contains a discussion of the findings; new findings cannot be present in the Discussion section.

The conclusions section has been modified to follow its indications, and eliminating that indicated part.

Tables, Figures

T2 replace <Estudios> by Educational level

We apologize for not having translated the term. It has already been modified.

Clarify <Piece with no tap> <Piece with a tap> <Pieces with no tap> <Pieces with a tap>; I don't understand

<in Table 3).>

As a suggestion to the reviewer's comment, we have modified table number 2 to clarify the concept

Why no Table 5?

The relationship between SC and QoL is represented in Table 4, in which the spearman correlation has been carried out between the Caespo (SC) dimensions and the STOMA QoL score. We have modified the table to make these questions much clearer.

Reviewer 4 Report

Dear Authors, i read with interest the Spanish multicentre crosssectional study about quality of life and stoma. I commend authors for compiling this important data that focus on QoL.

I have some comments for enhancing the discussion section:

  1. Lines 267-276 in discussion. It is important to discuss this matter of QoL negatively influenced by complications or morbidity. This is not new. This is both previously experiences and published and reported in oncology (e.g HCC) as well as benign gallbladder conditions (e.g. gallstones) and it needs to be alluded in discussion. Patients dont like complications to occur. So when complications occur (may not be negligence), the perception of own QoL as well as determinants of Quality of care provided - everything drops. Consider citing - Ahmed S, et al. Quality of Life in Hepatocellular Carcinoma Patients Treated with Transarterial Chemoembolization. HPB Surg. 2016;2016:6120143. doi: 10.1155/2016/6120143. Epub 2016 Apr 7. PMID: 27143815; PMCID: PMC4838811. as well as 
  2. In addition to above, patients do value QoL as an important metric and many patient reported outcome studies have concluded that QoL is integral to patient reported outcome measures. Consider citing - Mak MHW, et al. Patient reported outcomes in elective laparoscopic cholecystectomy. Ann Hepatobiliary Pancreat Surg. 2019 Feb;23(1):20-33. doi: 10.14701/ahbps.2019.23.1.20. Epub 2019 Feb 28. PMID: 30863804; PMCID: PMC6405362. The point here is important for both clinicians and patients to understand so a mutual common base exist for moderating expectations of outcomes.
  3. If data is available, it is good to know how many patients had chemo-radiation, chemotherapy, stage of disease, etc = some clinical data actually. It is well known that outcomes will affect survey. In addition, disease progression also affects QoL and other dimensions of the survey. So staging, progression etc some clinical data would be value add if available to authors, please report. 
  4. Before concluding section, authors must explicitly state strength and weakness of their study. I read some comparion with other national studies etc - but i need to see your study strengths (e.g. multi center, first time using CAESPO, sample size calculation etc etc) and weaknesses (excluding some pts, lack of clinical data, lack of clinical correlation with disease stage and progression or treatment response etc.)

Author Response

First of all, we want to acknowledge the contributions and comments of all reviewers. We believe that we have been able to take them all into account, which has substantially improved the article we present. Next, we will comment on each of the points indicated by your review.

1. Lines 267-276 in discussion. It is important to discuss this matter of QoL negatively influenced by complications or morbidity. This is not new. This is both previously experiences and published and reported in oncology (e.g HCC) as well as benign gallbladder conditions (e.g. gallstones) and it needs to be alluded in discussion. Patients dont like complications to occur. So when complications occur (may not be negligence), the perception of own QoL as well as determinants of Quality of care provided - everything drops. Consider citing - Ahmed S, et al. Quality of Life in Hepatocellular Carcinoma Patients Treated with Transarterial Chemoembolization. HPB Surg. 2016;2016:6120143. doi: 10.1155/2016/6120143. Epub 2016 Apr 7. PMID: 27143815; PMCID: PMC4838811. as well as 

2. In addition to above, patients do value QoL as an important metric and many patient reported outcome studies have concluded that QoL is integral to patient reported outcome measures. Consider citing - Mak MHW, et al. Patient reported outcomes in elective laparoscopic cholecystectomy. Ann Hepatobiliary Pancreat Surg. 2019 Feb;23(1):20-33. doi: 10.14701/ahbps.2019.23.1.20. Epub 2019 Feb 28. PMID: 30863804; PMCID: PMC6405362. The point here is important for both clinicians and patients to understand so a mutual common base exist for moderating expectations of outcomes.

Thank you very much for your suggestions, these are very interesting articles. Unfortunately, in order to integrate them into the content of the text, we would have to make a thorough reading and a series of modifications that, with the obligation to respond the comments of all the reviewers and the short time to carry out the modifications, it has been impossible for us to integrate them into the text. We will read them carefully and take them into account for future publications.

If data is available, it is good to know how many patients had chemo-radiation, chemotherapy, stage of disease, etc = some clinical data actually. It is well known that outcomes will affect survey. In addition, disease progression also affects QoL and other dimensions of the survey. So staging, progression etc some clinical data would be value add if available to authors, please report. 

The subjects participating in the research had an ostomy bag for a mean of 1780 days, with a standard deviation greater than 1000 days, that is, patients with long-term ostomy. The application of radiological and / or chemotherapeutic therapies is carried out postoperatively or in initial stages generally. So, collect this information 3 years later was not considered useful for this study, where was intended to assess self-care and quality of life of these patients with long-term ostomies.

Before concluding section, authors must explicitly state strength and weakness of their study. I read some comparion with other national studies etc - but i need to see your study strengths (e.g. multi center, first time using CAESPO, sample size calculation etc etc) and weaknesses (excluding some pts, lack of clinical data, lack of clinical correlation with disease stage and progression or treatment response etc.)

We have added the sections "Strenghts and Limitations" and "Implications" following your and other reviewers instructions.

Round 2

Reviewer 3 Report

please have a look at the attachment

Author Response

Thank you very much for your indications. All modifications made are marked in yellow.

Abstract

Methods

<Stoma Quality of Life (S-QoL)> and  (Results) <health-related quality of life (HRQoL)> - is there a difference between both, and if yes, what?

No, there is no difference. May be our presentation have been confusing. The dependent variable used in our study is the specific health-related quality of life (HRQOL) in patients with ostomy, measured through the Stoma Quality of Life (S-QoL) questionnaire.
We have clarified this concept by modifying the abstract.

<Trained interviewers> - what did these interviewers do, and in what did you train them?

As we indicated in the previous review: “The interviews were conducted by 3 different researchers. A common problem when using different interviewers is that each one has a style of writing or collecting information. To avoid this bias, a series of training sessions were carried out to homogenize their information collection styles, of course prior to the beginning of the research”.

All this is impossible to include in the abstract, due to space limitations. We add “to homogenize responses” in the abstract, and more information after Table 1, when we talk about the trained interviewers.

Were CAESPO and S-QoL already in the electronic medical record? It is not clear to me from the Abstract.

We add “that are not included in the electronic medical record” in the abstract.

Introduction

<your health related quality of life> - one’s (HRQoL)

Ups, our apologies. We did not understand the previous indication correctly. We have already made the modification.

To better compare the USA and Europe it would be better to show the numbers divided by an identical number (say 100,000)

We have homogenized the references, showing the numbers by 100,000 inhabitants.

Methods

Measures

<in the original study (32).> - how were these outcomes in your sample (thus not in the original study)?? And

Perhaps the expression in its current form is misleading, therefore we have proceeded to remove it. We were referring to the questionnaire validation article.

how did you translate this questionnaire? (you gave me an answer, but I think all of your readers will ask you this)

Tha CAESPO questionnaire has been created in Spanish and applied to native Spanish subjects. S-QoL have an Spanish adaptation and validation to Spanish (https://hqlo.biomedcentral.com/articles/10.1186/1477-7525-3-62) --> ref 32

Discussion

Please come with paras on S&l and Implications before the Conclusions

We have relocated those sections.
